# Ethyl Acetate Fraction of *Bixa orellana* and Its Component Ellagic Acid Exert Antibacterial and Anti-Inflammatory Properties against *Mycobacterium abscessus* subsp. *massiliense*

**DOI:** 10.3390/antibiotics11060817

**Published:** 2022-06-17

**Authors:** Roberval Nascimento Moraes-Neto, Gabrielle Guedes Coutinho, Ana Caroline Santos Ataíde, Aline de Oliveira Rezende, Camila Evangelista Carnib Nascimento, Rafaela Pontes de Albuquerque, Cláudia Quintino da Rocha, Adriana Sousa Rêgo, Maria do Socorro de Sousa Cartágenes, Ana Lúcia Abreu-Silva, Igor Victor Ferreira dos Santos, Cleydson Breno Rodrigues dos Santos, Rosane Nassar Meireles Guerra, Rachel Melo Ribeiro, Valério Monteiro-Neto, Eduardo Martins de Sousa, Rafael Cardoso Carvalho

**Affiliations:** 1Graduate Program in Health Sciences, Federal University of Maranhão—UFMA, São Luís 65080-805, MA, Brazil; roberval.moraes@discente.ufma.br (R.N.M.-N.); gabrielle.guedes@discente.ufma.br (G.G.C.); ana.ataide@discente.ufma.br (A.C.S.A.); aline.rezende@discente.ufma.br (A.d.O.R.); camila.carnib@ufma.br (C.E.C.N.); rafaela.pontes@hotmail.com (R.P.d.A.); cartagenes.maria@ufma.br (M.d.S.d.S.C.); anasilva1@professor.uema.br (A.L.A.-S.); rosane.guerra@ufma.br (R.N.M.G.); melo.rachel@ufma.br (R.M.R.); valerio.monteiro@ufma.br (V.M.-N.); eduardo.martins@ceuma.br (E.M.d.S.); 2Graduate Program in Chemistry, Federal University of Maranhão—UFMA, São Luís 65080-805, MA, Brazil; rocha.claudia@ufma.br; 3Graduate Program in Health and Services Management, CEUMA University—UniCEUMA, São Luís 65075-120, MA, Brazil; adriana.dsousa@ceuma.br; 4Graduate Program in Animal Science, University of Maranhão State—UEMA, São Luís 65055-310, MA, Brazil; 5Graduate Program in Biotechnology and Biodiversity Network BIODSNORTE, Federal University of Amapá—UFAP, Macapá 68903-419, AP, Brazil; igorsantosvictor@gmail.com (I.V.F.d.S.); breno@unifap.br (C.B.R.d.S.); 6Graduate Program in Microbial Biology, CEUMA University—UniCEUMA, São Luís 65075-120, MA, Brazil

**Keywords:** *Bixa orellana*, bioprospecting, in silico, *Tenebrio molitor*, anti-mycobacteria activity, anti-inflammatory activity

## Abstract

*Mycobacterium abscessus* subsp. *massiliense* (*Mabs*) causes chronic infections, which has led to the need for new antimycobacterial agents. In this study, we investigated the antimycobacterial and anti-inflammatory activities of the ethyl acetate fraction of *Bixa orellana* leaves (BoEA) and ellagic acid (ElAc). In silico analysis predicted that ElAc had low toxicity, was not mutagenic or carcinogenic, and had antimicrobial and anti-inflammatory activities. Apparently, ElAc can interact with COX2 and Dihydrofolate reductase (DHFR) enzymes, which could explain both activities. In vitro analysis showed that BoEA and ElAc exerted antimicrobial activity against *Mabs* (minimum inhibitory concentration of 1.56, 1.56 mg/mL and bactericidal concentration of 6.25, 3.12 mg/mL, respectively. Clarithromycin showed MIC and MBC of 1 and 6 µg/mL). Treatment with BoEA or ElAc increased survival of *Tenebrio molitor* larvae after lethal infection with *Mabs* and reduced carrageenan-induced paw edema in mice, around 40% of edema volume after the fourth hour, similarly to diclofenac. In conclusion, BoEA and ElAc exert antimicrobial effects against *Mabs* and have anti-inflammatory effects, making them potential sources of antimycobacterial drugs. The biological activities of ElAc may be due to its high binding affinities predicted for COX2 and DHFR enzymes.

## 1. Introduction

*Mycobacterium abscessus* (*Mabs*) is a nontuberculous mycobacterium that has recently emerged as the causative agent for a wide spectrum of clinical manifestations, including pulmonary infections, mainly in patients with cystic fibrosis (95% of cases) [1]. This fast-growing microorganism is one of the most important drug-resistant mycobacterial species, owing to its intrinsic resistance to several classes of antibiotics [2]. The low permeability of mycobacterial cell wall plays a major role in this intrinsic antibiotic resistance [3,4]. Dormancy and latency causing increased tolerance to antimicrobial agents that are lethal to replicating bacilli, a phenomenon referred to as phenotypic drug resistance, and efflux pumps that actively transport many antibiotics out of the cell and are major contributors to the intrinsic resistance of mycobacteria to many drugs. Furthermore, the intrinsic resistance to several important antibiotics in mycobacteria can also be observed by modification of antibiotics (enzymatic degradation of antibiotics) and modification of target sites, for example, erythromycin resistance methylase (*erm*) gene [5].

Patterns of genetic susceptibility can be used as predictors of antibiotic efficacy, thereby guiding the choice of antimicrobials to be used in *Mabs* infections [6]. The molecular mechanism associated with macrolide resistance is associated to the expression of the erythromycin ribosome methylase (*erm*) gene. Other resistance mechanisms include the *rrs* gene, which is associated with resistance to aminoglycosides, class A beta-lactamase associated with resistance to majority of the beta-lactams (except cefoxitin and imipenem), enzymatic inactivation of majority of the tetracyclines (except tigecycline) [7], and *gyrA* and *gyrB* genes, which are associated with quinolone resistance [6].

In this context, the use of natural products (phytochemical compounds) to treat conditions caused by resistant microorganisms can be advantageous because of their wide range of biological activities, which make them potential sources for the development of new drugs, including antimycobacterial agents [8,9,10,11]. *Bixa orellana* is a plant that possesses anti-inflammatory, antioxidant, antinociceptive, anticonvulsant, cardioprotective, and antidiabetic properties. This plant species has also been considered a promising source for antimicrobial agents against several microorganisms, given its various biological properties, ethnopharmacological relevance, and widespread distribution [12,13].

Our group previously characterized the ethyl acetate fraction obtained from *B. orellana* leaf hydroalcoholic extract (BoEA) [14,15] and identified ellagic acid (ElAc) as one of the active compounds. ElAc exerts several pharmacological activities that can potentially be used for the treatment and prevention of cancer, diabetes, and cardiovascular and neurodegenerative diseases. Additionally, it also has nephroprotective, hepatoprotective, antiviral, and antiparasitic properties [16,17]. Previous studies from our group demonstrated the antimicrobial activity of BoEA against Gram-positive bacteria, *Staphylococcus aureus* and *Streptococcus pyogenes* [18] and its anti-inflammatory activity was confirmed in vivo by reduction in total leukocytes migration to the peritoneum in mice [14].

In this research, specifically, it is important to highlight that the use of ellagic acid was performed in a bioguided study. After *B. orellana* leaf collection and processing with the preparation of the crude extract, a lyophilized compound called BOHE (Hydroalcoholic Extract of *B. orellana*) was prepared. Then, the BOHE was subjected to liquid–liquid fractionation with hexane, chloroform, and ethyl acetate, producing hexane (BoHex), chloroform (BoCl), and ethyl acetate (BoEA) fractions. The BoHex and BoCl fractions were stored to be used in further studies and, the ethyl acetate fraction (BoEa) was used in this study. The BoEA was subjected to column chromatography, which identified 10 groups or subfractions, and for each subfraction in vitro assays against *Mabs* were performed. The best results were shown by subfraction or group 10. Then, using HPLC-PDA and FIA-ESI-IT/MS it was characterized that the group/subfraction 10, is ellagic acid, the compound that we directed our study.

Considering the biological properties and antimicrobial potential of *B. orellana* as well as the clinical relevance of *Mabs*, this study aimed to evaluate the anti-inflammatory and antimicrobial activities of the ethyl acetate fraction of *B. orellana* and ElAc against the microorganism in silico, in vitro, and in vivo. In addition to providing a strategy for alternative therapies for mycobacterial infections, this study aimed to contribute to the development of new drug formulations.

## 2. Results

### 2.1. Identification of Phytochemical Compounds–Profile of Fraction Groups

A total of 10 fraction groups (G1–G10) of different compositions were obtained using classical column chromatography. The results of each fraction group, as well as the representative spectra for each peak obtained in the ultraviolet (UV) region, are presented in Table 1.

Group G10 was a pure compound, and showed the same spectrum in the UV region and retention time when compared with the ElAc standard by high-performance liquid chromatography with photodiode array detector (HPLC-PDA). Flow injection analysis-electrospray ionization-ion trap mass spectrometry (FIA-ESI-IT/MS) further confirmed that the isolated substance was ElAc (Figure 1 and Table 2).

From the HPLC–DAD analysis and the construction of the quantification curve, it was possible to measure that the concentration of ElAc in BoEA was 5.93%.

### 2.2. In Silico Analysis of the Biological Activities of ElAc and Prediction of Its Toxic Effects, Including Hepatotoxicity

The biological activity spectra of ElAc were determined using the prediction of activity spectra for substances (PASS) software. Table 3 shows the probable activity (Pa) and probable inactivity (Pi) values. Several activities of ElAc were predicted, including anti-inflammatory, antioxidant, antibacterial, antimycobacterial, and hepatoprotective activities. The highest Pa value was obtained for anti-inflammatory activity (0.749).

The predictive analysis of ADME-Tox properties was performed for the compounds clarithromycin, diclofenac and ellagic acid. Table 4 shows the results of the ADME parameters of PPB, hepatotoxic, solubility, BBBP, HIA and violations of Lipinski’s rule of five [19].

The prediction of toxicity through models of carcinogenicity in rodents (female mouse and rat female), Ames mutagenicity, skin irritancy, skin sensitization and ocular irritancy is shown in Table 5. The risk of toxicity and the prediction of carcinogenic potential are described in Table 6 and Table 7.

### 2.3. Molecular Docking

For docking studies, in an attempt to identify potential targets that could explain anti-inflammatory mechanisms of action and also the antimicrobial activity against *Mabs*, some targets were selected from the Protein Data Bank. Table 8 lists their data, including native ligands and grid center coordinates.

The comparison of the crystallographic ligands diclofenac, meclofenamic acid, MMV, Q0Q and JD8 (yellow color) and the best conformation predicted by molecular docking via DockThor [28] (green color), as well as the respective RMSD obtained in each analysis, can be seen in Figure 2.

The validation of molecular docking was accepted through the calculation of RMSD, where values in the range of 0.17–0.48 Å were observed (Figure 2). Then, a study of the interactions between native ligands/controls and selected targets, in their respective binding pockets, was performed. The types of interactions as well as their distances are shown in Appendix A.

After validation of molecular docking protocols, predictions of binding affinity (ΔG) and modes of interaction of native ligands, diclofenac (control used experimentally) and ellagic acid, were performed (Figure 3 and Appendix A). The murine COX2 enzyme presented binding affinities (∆G) of −9.1 and −8.2 (kcal/mol), with diclofenac and ellagic acid, respectively (Figure 3a). Predictions with the human COX2 also showed comparable binding affinities of −9.4, −9.3 and −9.1 (kcal/mol) with meclofenanic acid, diclofenac and ellagic acid, respectively (Figure 3b). Predictions by Dockthor showed that ellagic acid interacts with SER498 of the murine COX2 and with TYR322 and SER497 of the human COX2 enzyme (Appendix A).

Antimicobacterial activity was investigated using specific *Mabs* targets that were available in the Protein Data Bank, which included Dihydrofolate reductase (PDB ID 7K6C, DHFR), Phosphoribosylaminoimidazole-succinocarboxamide synthase (PDB ID 6YYB, SAICAR Synthetase), and tRNA (guanine-N(1)-)-methyltransferase (PDB ID 6QR4). Predictions of binding affinities showed that ellagic acid had binding affinity value that is similar (−8.3 kcal/mol) to that of the specific inhibitor (MMV) for the DHFR enzyme (Figure 4a).

Predictions of the modes of interaction of ellagic acid showed interactions with the DHFR enzyme through 5 conventional hydrogen bonds with ALA8, ASP28 and THR47 (Appendix A). Its interaction with the SAICAR Synthetase and methyltransferase (TrmD) targets also occurred through conventional hydrogen bonds (Appendix A); however, they resulted in lower binding affinities than that observed with DHFR (Figure 4).

The diagram of the interaction modes of the control compounds (diclofenac and clarithromycin) and the referred targets are presented in Appendix A.

### 2.4. Antibacterial Activity of BoEA and ElAc

BoEA showed a MIC of 1.56 mg/mL and MBC of 6.25 mg/mL, while ElAc showed a MIC of 1.56 mg/mL and MBC of 3.12 mg/mL. The Clarithromycin showed an MIC of 1 µg/mL and MBC of 6 µg/mL.

### 2.5. Time–Kill Assay

To evaluate the antimicrobial activity of BoEA and ElAc at different time intervals, a time–kill curve assay was performed over a period of 12–120 h with Mabs in the presence of various concentrations of BoEA and ElAc (MIC, 2 × MIC, and 4 × MIC). Table 9 shows that BoEA at 2 × MIC and 4 × MIC inhibited the growth of Mabs when compared to MIC and the control. On the other hand, ElAc considerably reduced the growth of Mabs compared to the control.

### 2.6. BoEA and ElAc Increase Survival of Tenebrio molitor Infected with Mabs

*Tenebrio molitor* larvae inoculated with *Mabs* were treated with BoEA and ElAc at doses corresponding to the MIC, 2 × MIC, and 4 × MIC. Treatment with BoEA or ElAc increased the survival of infected *T. molitor* compared to that of the untreated group (phosphate-buffered saline (PBS)-treated) (Figure 5). Both treatments showed a dose-dependent effect, as treatment with the highest concentration (4 × MIC) resulted in larval survival until day 8. In addition, the lowest dose of ElAc (MIC) was able to achieve 50% larval survival on day 10.

### 2.7. Anti-Inflammatory Activity of BoEA and ElAc

BoEA and ElAc showed anti-inflammatory effects on carrageenan-induced paw edema in mice (Figure 6). ElAc (50 mg/kg) acted similarly to diclofenac (15 mg/kg) and showed a long-lasting effect when compared to BoEA, as the inhibition could be observed until the third time.

## 3. Discussion

In the present study, we demonstrated through in silico, in vitro and in vivo analyses that the ethyl acetate fraction of *B. orellana* leaves (BoEA) and ellagic acid (ElAc) have both antimicrobial against *Mabs* and anti-inflammatory activity. ElAc was chosen for further studies after HPLC revealed its presence as one of the components of BoEA and because the literature suggests that the compound has many biological activities [16,17,18].

In silico prediction of the biological effects of ElAc revealed antioxidant, hepatoprotective, anti-inflammatory, and antibacterial activities. Zhao et al. [20] previously reported the hepatoprotective properties of ElAc, which reinforced the findings of our in silico prediction. They reported that mice with alcohol-induced liver disease treated orally with ElAc showed increased activity of alanine aminotransferase and serum aspartate aminotransferase, increased levels of triglyceride, low-density lipoprotein, free fatty acid, and total cholesterol, and decreased high-density lipoprotein levels. ElAc also showed antioxidant activities in the hepatic milieu, based on the levels of glutathione peroxidase, catalase, malondialdehyde, superoxide dismutase, and glutathione. Additionally, treatment with ElAc reduced proinflammatory cytokines, such as interleukin (IL)-6, IL-1β, and tumor necrosis factor-alpha (TNF-α), and the expression of several genes associated with the inflammatory immune response, including *TLR4*, *Myd88*, *CD14*, cyclooxygenase-2 [*COX2*], nitric oxide synthase 2 (*NOS2*), and nuclear factor kappa β (*NF-κ**β1*) [20]. The authors concluded that ElAc decreased oxidative stress, inflammatory response, and steatosis in mice with alcohol-induced liver disease.

Oxidative stress impairs antioxidant defenses (enzymatic and non-enzymatic), and the consequence of this imbalance is damage to major cellular components, which further leads to a gradual loss of tissue and organ function [29]. Consistent with our findings, Aslan et al. [30] demonstrated the antioxidant potential of ElAc in lung damage induced by carbon tetrachloride in rats. Rats with carbon tetrachloride-induced lung damage treated with ElAc showed reduced levels of inflammatory proteins NF-κβ and COX2, and proinflammatory cytokine TNF-α, establishing the protective role of ElAc in lung damage.

The parameters used to predict bioavailability of clarithromycin were used to predict bioavailability of ElAc in order to compare the results with previous reports. With regard to intestinal absorption, ElAc was found to be superior to clarithromycin, which suggests that in an in vivo study, a larger amount of clarithromycin would be needed to obtain the same serum concentration of ElAc [31]. Additionally, in silico analysis revealed that ElAc does not have mutagenic, tumorigenic, and irritating effects, or harmful effects on the reproductive system [32,33].

To validate the molecular docking methodology in our study, native ligands were submitted to redocking simulations, using the DockThor receptor–ligand docking program [28]. Gowthaman et al. [34] and Hevener et al. [35] reported that the molecular docking methodology will be validated when the RMSD calculated through redocking analysis—i.e., between the crystallographic pose and the computationally predicted pose is less than 2.00 Å. Then, based on the results shown in this analysis, the molecular docking analysis was validated for the five crystal structures used.

To evaluate the in silico anti-inflammatory activity of ElAc via COX_2_ inhibition, two crystal structures were selected: the crystal structure PDB ID 1PXX deposited by Rowlinson et al. [23] and PDB ID 5IKQ, deposited by Orlando and Malkowski [24]. The interactions predicted by Dockthor with TYR353 and SER498 are the same as those observed in the crystallographic pose, since during the protein preparation, the sequence is altered by the software itself. However, ellagic acid interacts with COX_2_ through its phenolic group linked to SER498, and it is possible to notice several interactions such as Pi-Sigma, Pi-Alkyl, Carbon Hydrogen Bond, and Amide Pi-Stacked, between amino acid residues and the atoms of its aromatic rings.

Docking studies carried out to identify possible targets related to inhibitory activity against *Mabs* revealed that ElAc has a binding affinity value (-8.3 kcal/mol) equal to that presented by the specific inhibitor (MMV) for DHFR, suggesting that its antibacterial effect may be linked to the inhibition of this enzyme than to the other two targets analyzed (SAICAR Synthetase and tRNA methyltransferase), since the predictions revealed that ElAc has lower binding affinities to these latter structures, when compared to their native ligands. DHFR is an enzyme with an important role in the folic acid pathway, participating in the synthesis of nucleic acid precursors, among other functions related to an improvement in DNA translation, RNA transcription and protein replication and, thus, controlling bacterial multiplication [36]. Therefore, DHFR inhibitor compounds can lead to bacterial death. We cannot yet be assured of whether ellagic acid can be considered an inhibitor of this enzyme, but our in silico data show considerable binding affinity indicating this enzyme as a possible target that needs further investigation.

Regarding the antibacterial effect, we observed that BoEA and ElAc exerted bactericidal effects. The antimicrobial activity of BoEA has been reported previously against various microorganisms, such as Gram-positive bacteria (*S. aureus* and *S. pyogenes*) [18], Gram-negative bacteria *(Escherichia coli*, *Salmonella typhi*, *Shigella dysenteriae*, *Klebsiella pneumoniae*, *Proteus vulgaris*) [37], yeast (*Candida albicans*) [38], antileishmanial [39], and antimalarial activities [40]. Our group previously demonstrated that BoEA has antimycobacterial activity against *Mabs* [14]. In addition, in the present study, we provide evidence for the antimycobacterial activity of ElAc, which was isolated from BoEA.

The antimycobacterial activity of BoEA can be explained by the chemical characteristics of its constituent polar compounds. Compounds with intermediate to high polarity inhibit the growth of mycobacterial species [41]. Although we could not confirm the exact phytochemicals responsible for the observed antimycobacterial activity against *Mabs*, it is highly possible that the observed activity was due to the presence of ElAc in BoEA. We used various approaches including in silico, in vitro, and in vivo analyses to confirm that the phytochemical ElAc from BoEA demonstrated antimicrobial activity against *Mabs*.

Other studies have evaluated the antimycobacterial activity of ElAc isolated from other plants. Dey et al. [42] reported that ElAc from pomegranate fruit exerts antimycobacterial activity against *Mycobacterium tuberculosis* (MIC 64–512 µg/mL). Sridevi et al. [43] reported that ElAc from pomegranate peels inhibited *M. tuberculosis* with an MIC of 0.3–3.5 mg/mL. Salih et al. [44] reported that ElAc from *Combretum hartmannianum* exerts antimycobacterial activity against *Mycobacterium smegmatis* (MIC 500–1000 μg/mL) and in another study, the same group [45] reported that ElAc from *Anogeissus leiocarpa* exhibited growth inhibitory activity against *M. smegmatis* with an MIC value of 500 µg/mL. However, to the best of our knowledge, we are the first to report the biological activity of ElAc from *B. orellana* leaves against *Mabs*. The MIC and MBC values of ElAc against *Mabs* in the present study were 1.56–3.12 mg/mL. These values are higher than those reported for E1Ac in previous studies. This can be explained by the intrinsic resistance of *Mabs* compared with that of other mycobacterial species. Treatment of *M. abscessus* infection is challenging due to the high level of innate resistance of the bacteria and is often associated with lengthy, costly, and non-standardized administration of antimicrobial agents. Moreover, adverse effects, drug toxicities, and high relapse rates result in poor treatment outcomes [46].

In the present study, the antimicrobial effects of BoEA and ElAc were tested in *Mabs*-infected *T. molitor* larvae. We observed that BoEA and ElAc at MIC, 2× MIC, and 4× MC inhibited *Mabs* infection in *T. molitor* larvae [47]. Larvae treated with BoEA and ElAc exhibited greater survival than untreated larvae, emphasizing the protective effect of these compounds against *Mabs* infection. The ability of *T. molitor* to produce reactive species in response to deleterious stimuli makes it a potential model for studying antimicrobial substances. Despite these advantages, to the best of our knowledge, no study has used *T. molitor* larvae for screening plant-derived antioxidant compounds against *Mabs.* This approach could provide more information than models traditionally used for studying antimicrobial agents, based on the chemical interaction of compounds and without biological activity relevance [48].

To evaluate the anti-inflammatory activity of BoEA and E1Ac, we measured the inhibition rate of carrageenan-induced paw edema in mice. Inflammation limits the damage to cells after invasion by foreign organisms or mechanical injury. Histamine is a common inflammatory mediator [49]. Inflammatory responses occur rapidly and eventually lead to vasodilatation and plasma exudation, which induces recruitment of inflammatory mediators and causes edema [50]. Reduction in carrageenan-induced acute inflammation and the resulting paw edema is a useful marker for investigating the anti-inflammatory potential of drugs and plant extracts [51].

Development of carrageenan-induced edema occurs in two phases: the initial phase involves histamine and serotonin, and the later phase is characterized by a marked increase in prostaglandin production [52]. The results obtained in the present study suggest that the anti-inflammatory effect of BoEA on carrageenan-induced edema in mice affects both the early and late phases, observed at doses of 50 mg/kg and 150 mg/kg, respectively, where its bioactive components, including ElAc, possibly act by suppressing the inflammatory response mediated by histamine, serotonin, and/or prostaglandins, presenting a potential source for cyclooxygenase inhibitors [53]. Previously, it was reported that the anti-inflammatory mechanisms of ElAc were related to a decrease in the level of COX_2_ via the suppression of proinflammatory cytokines (TNF-α, IL-1β), NO, and Prostaglandin E2 overproduction [54].

Some limitations can be highlighted in our study, including that we evaluated the antimicrobial activity against only one isolate of *M. abscessus* subsp. *massaliense*. Although we did not include a reference strain in the study, the *Mabs* used in our study (strain G01) is a well-characterized and relevant clinical isolate, since it was cultured from 18 patients in an outbreak of post-surgical infections in a hospital in the city of Goiânia (Brazil) [55]. A second limitation concerns the small number of potential targets for antimicrobial activity that we analyzed. However, at the time of writing, the Protein Data Bank only has these three crystallographic structures obtained from *Mabs* with their respective inhibitors, which restricted our analysis.

## 4. Materials and Methods

### 4.1. Drugs, Chemicals, and Reagents

Hexane, ethanol, methanol, chloroform, acetic acid, dimethyl sulfoxide (DMSO), PBS, resazurin, clarithromycin, carrageenan, and ElAc were purchased from Sigma–Aldrich (St. Louis, MO, USA). Mueller-Hinton broth was purchased from Merck KGaA (Darmstadt, Germany). Middlebrook 7H11 agar base was purchased from HiMedia (Mumbai, India). Diclofenac sodium was purchased from Novartis Biociências S.A. (São Paulo, Brazil).

### 4.2. Preparation of BoEA and Isolation/Identification of ElAc

#### 4.2.1. Collection of *B. orellana* Leaves and Separation of Fractions

*B. orellana* leaves were collected in the municipality of São José de Ribamar, Maranhão, Brazil (April–May 2019) and identified at the herbarium of the Federal University of Maranhão (São Luís, Brazil, specimen voucher 1147, SISGen AE80277). The leaves were dried in an oven at 40 °C for three days and then dried at room temperature (24–26 °C) for another four days. The leaves were then ground in a mill and their crude extract was extracted by maceration in 70% ethanol for 24 h.

The sample was filtered, and the resulting filtrate was concentrated on a rotary evaporator under low pressure at 50 °C. The concentrate was lyophilized and labeled BoHE. BoHE was then subjected to liquid–liquid fractionation using hexane, chloroform, and ethyl acetate with a series of increasing polarities to produce hexane (BoHex), chloroform (BoCl), and ethyl acetate (BoEA) fractions. BoEA was used in the present study and the other fractions were stored for future use.

#### 4.2.2. Isolation and Characterization of Compounds in BoEA

The phytochemical compounds in BoEA were purified using a chromatography column (230–400 mesh; 8 × 100 cm) and eluted with increasing polarities of mixtures of n-hexane/ethyl acetate and ethyl acetate/methanol to obtain subfractions. The subfractions (395) were then grouped using thin-layer chromatography and HPLC-PDA. Ten fractions were obtained and analyzed using HPLC-PDA. Group 10 showed a single peak in the chromatogram, suggesting that it may have been a single compound.

The structure of group 10 was determined using HPLC-PDA and FIA-ESI-IT/MS. Mass spectrometry was performed using an LCQ Fleet Equipment (Thermo Fisher Scientific, Waltham, MA, USA) equipped with a device for direct sample insertion via FIA. The studied matrix was analyzed by ESI and multi-stage fragmentation (MS2, MS3, and MSn) was performed in an IT interface. The negative and positive modes were selected for the generation and analysis of mass spectra for the first order (MS) and the other multi-stage experiments were performed under the following conditions: capillary voltage of 25 V, spray voltage of 5 kV, and capillary temperature of 275 °C. A carrier gas (N_2_) with a flux of eight arbitrary units (was used and the collision gas was helium (He). The range of acquisition was *m*/*z* 100–1000. Xcalibur software version 1.3 (Thermo Fisher Scientific) was used to acquire and process the data.

### 4.3. In Silico Analysis

#### 4.3.1. Prediction of Biological Activities of ElAc

The biological activities of ElAc and clarithromycin (standard drug) were evaluated using the PASS Online platform (version 2.0, Way2Drug.com©2011–2022, Moscow, Russia) (www.way2drug.com/passonline/, accessed on 15 October 2021), which can predict several biological characteristics of a substance. The PASS program describes biological activity as “active” (Pa) or “inactive” (Pi), where the estimated probability varies from 0 to 1. The chances of finding a particular activity increase when the Pa values are higher and Pi values are lower. The results of PASS prediction were interpreted as follows: (i) only biological activities with Pa > Pi were considered probable for a particular compound; (ii) if Pa > 0.7, the substance is likely to exhibit the said biological activity and the probability of the compound being an analog of a known pharmaceutical drug is also high; (iii) if 0.5 < Pa < 0.7, the compound is likely to exhibit the said biological activity, but the compound is not similar to known drugs; (iv) if Pa < 0.5, the compound likely does not exhibit the said biological activity and is perhaps a structurally new compound [56].

#### 4.3.2. Prediction of Pharmacokinetic Characteristics and Toxic Effects of ElAc

Predictive values of absorption, distribution, metabolism, excretion, and toxicity (ADME-Tox) properties were calculated using Discovery Studio Client v16 software 1.0.15350 [21], following the methodological proposal described by Shukla et al. [57] and Ramos et al. [21]. The ADMET Descriptors protocol, run by the aforementioned software, uses a variety of QSAR models to estimate pharmacokinetic/toxicological properties for small molecules [22]. Thus, the properties analyzed were: binding to plasma proteins (PBB), hepatotoxicity, penetration of the blood brain barrier (BBBP), solubility and human intestinal absorption (HIA). Another important step for the ADME-Tox analysis is to evaluate possible violations of Lipinski’s Rule of Fives [19]. To carry out this methodological step, the “calculate molecular properties” module present in the Discovery Studio Client v16 software was used 1.0.15350 [22].

Toxicity predictions were performed using TOPKAT (Toxicity Prediction function by Komputer Assisted Technology) [19]. Therefore, such a module can predict the toxicity of chemicals based solely on their 2D molecular structure, using a variety of robust, cross-validated quantitative structure-toxicity relationship (QSTR) models to assess specific toxicological parameters. Therefore, the toxicological properties analyzed were: carcinogenicity in rodents (female mouse and rat female), Ames mutagenicity, skin irritancy, and skin sensitization. Toxicity risk prediction calculations were performed via TOPKAT and measured the following parameters: rate oral LD50 (g/kg BodyWeight), Daphnia EC50 (mg/L), rat chronic LOAEL (g/kg BodyWeight), fathead minnow LC50 (g/L). In addition, the carcinogenic potential was also predicted through the parameters: TD50 (mg/kg body weight/day-mouse/rat) and RMTD (Rat Maximum Tolerated Dose-mg/kg body weight).

#### 4.3.3. Molecular Docking Study

To investigate the anti-inflammatory activity by the inhibition of COX2 (Prostaglandin G/H synthase 2) two crystal structures were selected according to Rowlinson et al. [23] and Orlando et al. [24]. Furthermore, to evaluate the multi-target antibacterial effect, three crystal structures of the following enzymes were selected: dihydrofolate reductase (DHFR), SAICAR Synthetase (PurC), and tRNA (m 1 G37) methyltransferase (TrmD) [25,26,27]. To carry out the molecular docking study, it was necessary to preprocess the structures with the help of Discovery Studio Client software v16. 1. 0. 15350 [22], which allowed the exclusion of water molecules, metals, cofactors, separation of ligands and protein structures. Then, the DockThor software, a receiver–ligand docking program, was used (https://dockthor.lncc.br/v2/, accessed on 30 May 2022) [28]. The parameters used were: number of evaluations “1,000,000”, population size “750”, initial seed “−1985” and number of runs “24”. In addition, hydrogen was added (pH 7.0) by the software’s own module. The grid center was defined using Discovery Studio Client software [22], and the grid size and discretization were kept with their default values presented by the software, i.e., (X = 20, Y = 20 and Z = 20) and 0.25, respectively. The validation of each molecular docking analysis was performed through redocking simulations. This procedure was performed with the submission of the crystallized native ligand itself to DockThor [28]. Then, the RMSD (root-mean-square deviation) was calculated using the Discovery Studio Client software, using the crystallographic ligand pose and the computationally generated pose, using all atoms. In addition, comparative poses were generated. Finally, the analysis of interactions between the respective crystallized ligand and the enzymes in each binding pocket was performed, generating 2D diagrams using the Discovery Studio Client software.

The control compounds (diclofenac and clarithromycin) and the studied compound ellagic acid were submitted to molecular docking analysis with the parameters already described (Appendix A). In addition, the binding affinity (ΔG) results predicted by DockThor as a score (−kcal/mol) were plotted using the GraphPadPrism 7.0 software (GraphPad Software Inc., San Diego, CA, USA). Finally, to qualitatively analyze the interactions between the compounds and the aforementioned enzymes, 2D diagrams were generated in the respective binding pocket, using the Discovery Studio Client software.

### 4.4. In Vitro Analysis

#### 4.4.1. Strain

In the present study, we used the *Mabs* GO01 strain, which was cryopreserved (−80 °C) in the Laboratory of Immunology and Microbiology of Respiratory Infections of the CEUMA University, São Luís, Brazil. This isolate was collected from a patient with infections caused by contaminated instruments, following invasive procedures [55].

The use of the clinical isolate was authorized through a written informed consent form approved by the Certificate of Ethics Appraisal Presentation (CAAE:21357413.4.0000.5084).

#### 4.4.2. Determining MIC and MBC

MIC determination was performed based on the standard recommendations [58], using the Mueller Hinton broth microdilution technique in a 96-well plate, with different concentrations of clarithromycin (positive control; 0.001–1 mg/mL), BoEA (0.1–12.5 mg/mL), and ElAc (0.1–12.5 mg/mL). *Mabs* culture without any additive was used as the negative culture and was set as 100% growth (i.e., 0% inhibition). The experiment was performed in triplicate. On the third day of incubation, 30 µL of 0.01% resazurin was added to each well, and the plate was further incubated overnight. A change in color from blue (oxidized state) to pink (reduced state) indicates bacterial growth [59]; MIC value was defined as the lowest drug concentration that prevented bacterial growth.

The concentrations from the MIC determination assay that did not show visible growth were inoculated on a 7H11 agar plate for MBC determination. MBC was defined as the lowest concentration of the BoEA and E1Ac that inhibited mycobacterial growth (≥99.9% cell death) [60].

#### 4.4.3. Time–Kill Assay

Individual bottles with 20 mL of Middlebrook 7H9 with OADC growth supplement and 0.05% Tween 80 were inoculated with *Mabs* (10^5^–10^6^ CFU/mL) and BoEA or ElAc at MIC, 2× MIC, and 4× MIC was added to the bottles, which were then cultured at 37 °C. All bottles were shaken (100 rpm) and ventilated through a bacterial filter (FP 30/0.2 Ca/S, Whatman GmbH, Dassel, Germany). To perform CFU counting, 200 μL was taken from each bottle and serial 10-fold dilutions in 0.85% sterile saline solution were plated on Middlebrook 7H11 plates (BD Bioscience, Franklin Lakes, NJ, USA) at different time intervals (12, 24, 36, 48, 72, 96, and 120 h). Experiments were performed thrice in triplicate. The results are expressed as Log_10_ CFU.

### 4.5. In Vivo Analysis

#### 4.5.1. *T. molitor* Survival Assay

*T. molitor* larvae were fed ad libitum with bran flour and water supplemented with apples. For experiments, 10th–12th instar larvae without evident body injuries or melanization were used. The larvae were infected with 10 µL of *Mabs* (1 × 10^5^ CFU) using a Hamilton syringe equipped with a 26-gauge needle. They were injected intrahemocele, at the second or third sternite, above the legs, in the ventral portion. After inoculation, the larvae were kept in petri dishes at 37 °C, which contained chopped apples to avoid dehydration. Silk on the larval surface was removed as soon as possible to delay transition to the pupal stage.

Each experimental group was composed of 10 larvae, and they were distributed as follows: PBS group, negative control, not infected or treated, inoculated with 10 µL of sterile PBS; positive control group, infected with *Mabs* (1 × 10^5^ CFU) and treated with clarithromycin (10 µL/mL); *Mabs* group, infected with *Mabs* (1 × 10^5^ CFU), not treated; BoEA group, infected with *Mabs* (1 × 10^5^ CFU) and treated with BoEA at MIC, 2 × MIC, and 4 × MIC; and ElAc group, infected with *Mabs* (1 × 10^5^ CFU) and treated with ElAc at MIC, 2 × MIC, and 4 × MIC. The larvae were treated with clarithromycin, BoEA, or ElAc 1 h after inoculation with *Mabs* were monitored daily for 10 days. Dead animals were defined as those showing no signs of irritability, extensive body melanization, or shrinking [61].

#### 4.5.2. Carrageenan-Induced Paw Edema Model

C57BL/6 females, 6–8 weeks of age and weighing between 18 and 22 g, were procured from the Central Vivarium of CEUMA University. Mice were housed in an upper cage and allowed ad libitum food and water. The protocol for animal experiments was approved by the Ethics Committee on the Use of Animals, CEUA/UniCEUMA (protocol number 147/18). All animal experiments were performed according to the guidelines of the National Council for the Control of Animal Experimentation (CONCEA), Ministry of Science, Technology and Innovation (MCTI), Brazil, and the Brazilian Society of Science in Laboratory Animals (SBCAL).

Paw edema was induced by injecting 0.05 mL of 1% (*w*/*v*) carrageenan suspended in saline into the subplantar tissue of the right hind paw of each mouse [62]. A digital caliper (ZAAS Amatools, São Paulo, Brazil) was used to measure the paw thickness (Ct). An increase in paw thickness was considered an indicator of inflammation. Paw thickness was measured at various time points: at 0 h (C0), i.e., immediately after edema induction, and at 1, 2, 3, and 4 h after paw edema induction. All animals were pretreated by gavage 1 h before the induction of paw edema.

Mice were divided into six groups (n = 4 each): control group, paw edema was induced, and animals were treated with vehicle (10 mg/kg); BoEA groups, paw edema was induced and animals were treated with 25, 50, or 150 mg/kg of BoEA; ElAc group, paw edema was induced and animals were treated with 50 mg/kg of ElAc; and diclofenac group, paw edema was induced and animals were treated with 15 mg/kg.

The following equation was used to calculate percentage inhibition of inflammation [63]:$$\%\mathrm{Inhibition}\;=\frac{\left(\mathrm{Ct}-C0\right)\mathrm{Control}-\left(\mathrm{Ct}-C0\right)\mathrm{treated}}{\left(\mathrm{Ct}-C0\right)\mathrm{Control}}\times 100$$

Legend: Ct, paw thickness after treatment with carrageenan at time “t”; C0, initial (basal) paw thickness.

### 4.6. Statistical Analysis

Data are presented as the mean ± standard variation or percentages. The normality of distributions was determined using the Shapiro–Wilk test, and differences between groups were evaluated by analysis of variance followed by Tukey’s multiple comparison test using GraphPad Prism software (version 6.0, GraphPad Prism software Inc., San Diego, CA, USA). Statistical significance was set at *p* < 0.05. Larval survival assays were analyzed using the Kaplan–Meier method to calculate survival fractions, and the log-rank test was used to compare survival curves.

## 5. Conclusions

Our in silico, in vitro, and in vivo data indicate that BoEA and ElAc have anti-inflammatory and antimicrobial activity against *Mabs*. Apparently, the mechanisms of action are related to the binding affinity of ellagic acid to both COX_2_ and DHFR, respectively, which warranted further studies. Our results suggest that ElAc can be optimized to develop new lead compounds against antimicrobial resistant pathogens, such *Mabs* and/or anti-inflammatory drugs.

## Figures and Tables

**Figure 1 antibiotics-11-00817-f001:**
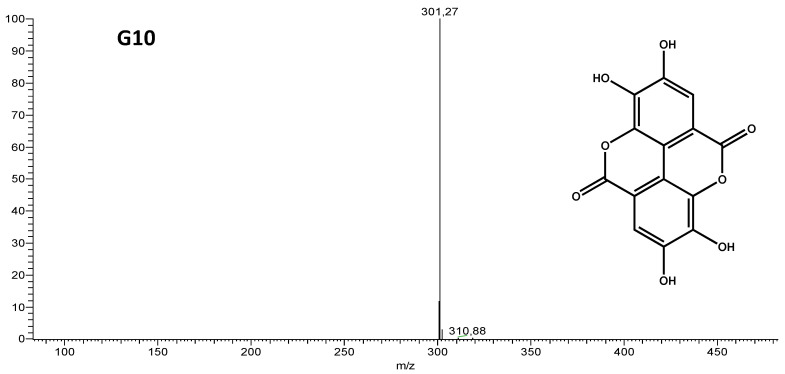
Mass spectrum of Group 10, obtained by flow injection analysis-electrospray ionization-ion trap mass spectrometry (FIA-ESI-IT/MS), ionization by negative mode and structure of the isolated compound (Ellagic acid).

**Figure 2 antibiotics-11-00817-f002:**
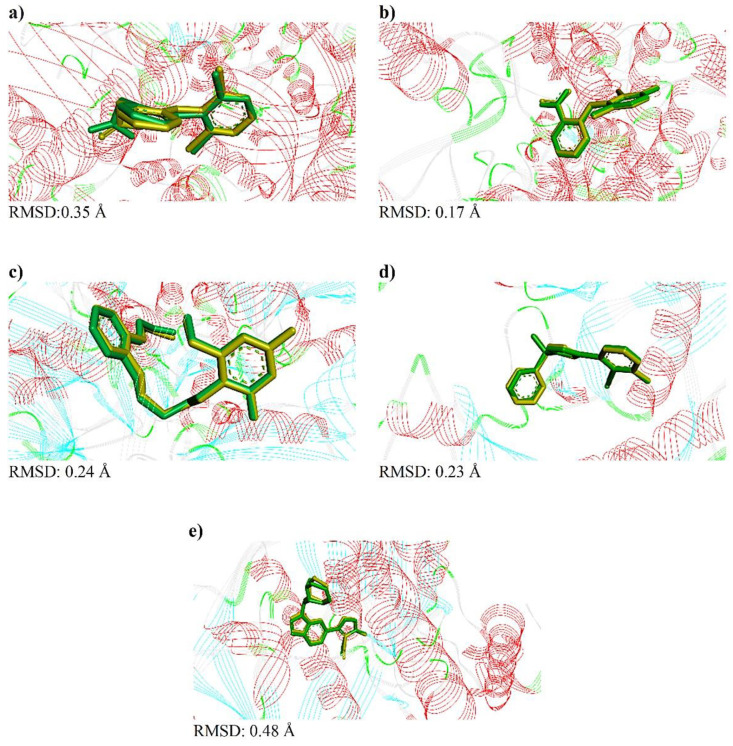
Redock results via Dockthor for ligands: (**a**) diclofenac, (**b**) meclofenamic acid, (**c**) MMV, (**d**) Q0Q e, (**e**) JD8. The yellow ligand corresponds to the crystallographic pose, while the green ligand corresponds to the computationally generated pose. The RMSD values for each analysis are shown below each image.

**Figure 3 antibiotics-11-00817-f003:**
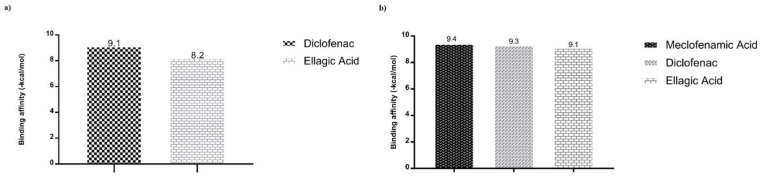
Binding affinity values of the ligands with the inflammation targets obtained by molecular docking via Dockthor: (**a**) with the murine COX2 (PDB ID 1PXX); and (**b**) with the human COX2 (PDB ID 5KIQ).

**Figure 4 antibiotics-11-00817-f004:**
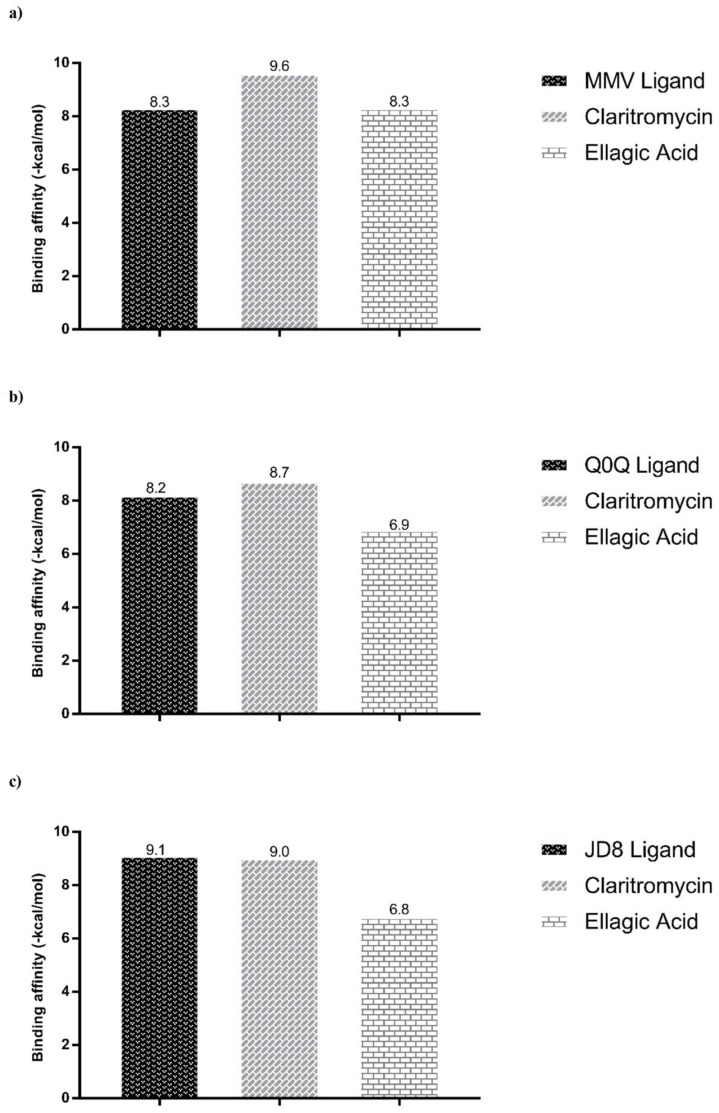
Binding affinity values of the native ligands, clarithromycin, and ellagic acid with targets of antimicrobial activity obtained by molecular docking via Dockthor. The selected targets were: (**a**) Dihydrofolate reductase (PDB ID 7K6C); (**b**) Phosphoribosylaminoimidazole-succinocarboxamide synthase (PDB ID 6YYB), and (**c**) tRNA (guanine-N(1)-)-methyltransferase (PDB ID 6QR4).

**Figure 5 antibiotics-11-00817-f005:**
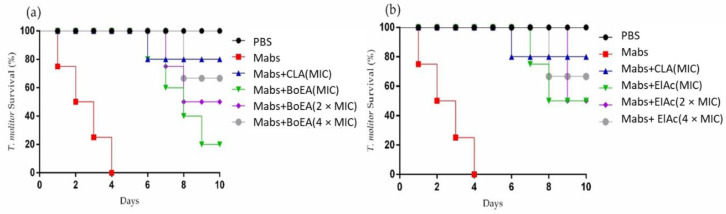
Effects of BoEA and ElAc on the *T. molitor* survival after lethal infection with *Mabs*. Effect of MIC, 2 × MIC, and 4 × MIC of (**a**) BoEA and (**b**) ElAc on larval survival. Ten larvae were included in each group and were monitored daily to assess mortality. PBS-treated group represents the negative control, while the clarithromycin-treated group represent the positive control. The results are representative of one of three independent experiments. Abbreviations: PBS, phosphate-buffered saline; CLA, clarithromycin; MIC, minimum inhibitory concentration; BoEA, ethyl acetate fraction of *B. orellana*; ElAc, ellagic acid.

**Figure 6 antibiotics-11-00817-f006:**
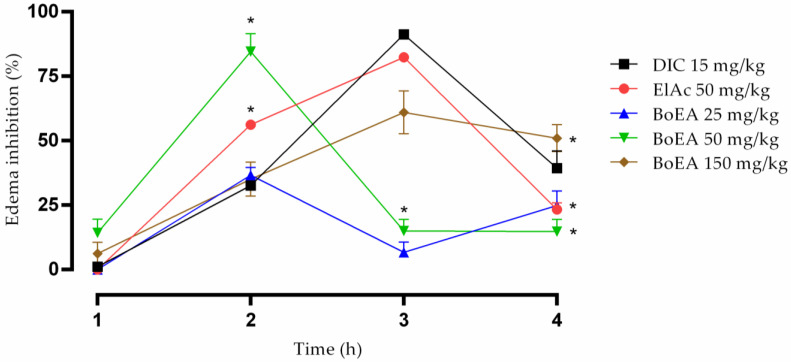
Anti-inflammatory effect of BoEA and ElAc in carrageenan-induced paw edema in mice. The thickness of the paw was measured 1, 2, 3, and 4 h after carrageenan injection. Data are presented as the mean ± SD. * *p* < 0.05 compared with the diclofenac-treated group. The results are representative of one of three independent experiments. Abbreviations: BoEA, ethyl acetate fraction of *B. orellana*; ElAc, ellagic acid; DIC, diclofenac.

**Table 1 antibiotics-11-00817-t001:** Data taken from high-performance liquid chromatography with photodiode array detector analyses of fractions obtained from *Bixa orellana* fractions.

Group	UV Max (nm)
G1 (mixture)	300; 325; 278
G2 (mixture)	300; 325; 278
G3 (mixture)	300; 325; 278
G4 (mixture)	300; 325; 278
G5 (mixture)	255; 267; 349
G6 (mixture)	253; 267; 345
G7 (mixture)	253; 267; 345
G8 (mixture)	253; 267; 345
G9 (mixture)	300; 325; 278
G10 (pure compound)	254; 300; 366

**Table 2 antibiotics-11-00817-t002:** Chemical characterization of G10 by FIA-ESI-MS.

[M-H]-	MS^n^ions	Identification
301	257 [M-44-H]-	Ellagic Acid
	229 [M-44-28-H]-	

Abbreviations: [M-H]-, deprotonated molecule; MS^n^, multi-stage mass.

**Table 3 antibiotics-11-00817-t003:** In silico analysis of the biological activities of Ellagic Acid (ElAc).

Activities	PASS Predictions of Ellagic Acid	
	Pa	Pi
Anti-inflammatory	0.749	0.010
Antioxidant	0.699	0.004
Antibacterial	0.380	0.035
Antimycobacterial	0.301	0.080
Hepatoprotective	0.599	0.012
DNA ligase (ATP) inhibitor	0.466	0.009
Superoxide dismutase inhibitor	0.423	0.058
DNA-3-methyladenine glycosylase I inhibitor	0.379	0.029
RNA directed DNA polymerase inhibitor	0.323	0.026
DNA polymerase I inhibitor	0.330	0.041
Nitric oxide scavenger	0.282	0.011
DNA synthesis inhibitor	0.261	0.074
Antibiotic	0.158	0.044
Antibacterial Ophthalmic	0.154	0.051
Cell wall synthesis inhibitor	0.094	0.072
tRNA-pseudouridine synthase I inhibitor	0.617	0.014

Abbreviations: Pa, probable activity; Pi, probable inactivity; PASS, prediction of activity spectra for substances.

**Table 4 antibiotics-11-00817-t004:** ADME prediction of controls and Ellagic Acid.

Compound	PPB	Hepatotoxic	Solubility	BBBP	HIA	Lipinski Violations (Max 4)
Clarithromycin	false	True	2	4	3	3
Diclofenac	true	True	2	1	0	0
Ellagic Acid	true	True	3	4	1	0

Abbreviations: PPB = plasma–protein binding (false: does not bind to plasma proteins, true: binds to plasma proteins); Hepatotoxic (true: hepatotoxic effect, false: no hepatotoxic effect); Aqueous solubility: (acceptable range: range is 0–3, where 3 is a good solubility) [20,21]; BBBP = 0 (very high penetrant), 1 (high), 2 (medium), 3 (low), 4 (undefined/very low), HIA: 0 (good); 1 (moderate), 2 (poor), 3 (very poor) [22].

**Table 5 antibiotics-11-00817-t005:** Topkat models (Mouse Female, Rat Female, Ames Mutagenicity, Skin Irritancy, Skin Sensitization and Ocular Irritancy) of clarithromycin, diclofenac and Ellagic Acid.

Compound	Mouse Female	Rat Female	Ames Mutagenicity	Skin Irritancy	Skin Sensitization	Ocular Irritancy
Clarithromycin	NC	NC	NM	Mild	None	Mild
Diclofenac	NC	NC	NM	None	Strong	Mild
Ellagic Acid	NC	NC	NM	None	Strong	Mild

Abbreviations: NC: non-carcinogen; NM: non-mutagen.

**Table 6 antibiotics-11-00817-t006:** Toxicity risk prediction via TOPKAT (Rate oral LD50, Daphnia EC50, rat chronic LOAEL, fathead minnow LC50).

Compound	Rat Oral LD50 (g/kg Body Weight)	Rat Chronic LOAEL (g/kg Body Weight)	Fathead Minnow LC50 (g/L)	*Daphnia* EC50 (mg/L)
Clarithromycin	0.787	0.001	0.004	0.278
Diclofenac	0.564	0.034	0.002	9.665
Ellagic Acid	0.428	0.302	0.049	16.407

Abbreviations: Daphnia EC50 = the effect concentration of a substance that causes adverse effects on 50% of the test population (*Daphnia magna*); LOAEL—lowest observed adverse effect level; Fathead minnow—short-term toxicity to fish.

**Table 7 antibiotics-11-00817-t007:** Carcinogenic potential via TOPKAT (Rat TD50 and RMTD).

Compound	Rat TD50 (mg/kg Body Weight/Day)	RMTD—Feed (mg/kg Body Weight/Day)
Clarithromycin	0.002	9.995
Diclofenac	51.448	742.520
Ellagic Acid	5.838	721.906

Abbreviations: RMTD: rat maximum tolerated dose; TD50: tumorigenic dose rate.

**Table 8 antibiotics-11-00817-t008:** Description of receptors related to antibacterial and anti-inflammatory activity used in the study of molecular docking.

Enzyme	Ligands Ligand ID/(Synonyms)	Coordinates of Grid Center
COX2 (Prostaglandin G/H synthase 2) (*Mus musculus*) PDB ID: 1PXX Resolution: 2.90 Å Reference: [23]	2-[2-[(2,6-dichlorophenyl)amino]phenyl]ethanoic acid DIF Diclofenac	X = 27,115 Y = 24,090 Z = 14,936
COX2 Prostaglandin G/H synthase 2 (*Homo sapiens*) PDB ID: 5IKQ Resolution: 2.41 Å Reference: [24]	2-[(2,6-dichloro-3-methyl-phenyl)amino]benzoic acid 2- / JMS Meclofenamic Acid	X = 21,597 Y = 51,876 Z = 17,696
Dihydrofolate reductase (*Mycobacteroides abscessus* ATCC 19977) PDB ID: 7K6C Resolution: 2.00 Å Reference: [25]	3-(2-{3-[(2,4-diamino-6-ethylpyrimidin-5-yl)oxy]propoxy}phenyl)propanoic acid/ MMV	X = -33,882 Y = −7502 Z = 56,281
Phosphoribosylaminoimidazole-succinocarboxamide synthase (*Mycobacteroides abscessus* ATCC 19977) PDB ID: 6YYB Resolution: 1.51 Å Reference: [26]	4-azanyl-6-[1-[(1~{R})-1-phenylethyl]pyrazol-4-yl]pyrimidine-5-carbonitrile Q0Q	X = 21,1853 Y = 14,726 Z = 34,7921
tRNA (guanine-N(1)-)-methyltransferase (*Mycobacteroides abscessus*) PDB ID: 6QR4 Resolution: 1.52 Å Reference: [27]	zanyl-3-[1-[[(2~{R})-1-methylpiperidin-2-yl]methyl]indol-6-yl]-1~{H}-pyrazole-4-carbonitrile JD8	X = -12,718 Y = 7688 Z = −27,062

**Table 9 antibiotics-11-00817-t009:** Time–kill evaluation of Mabs with BoEA and ElAc. Clarithromycin concentration was used at MIC. Growth control: no compound was added to the cell suspension.

Time (h)	Control Group	Antibacterial Agents						
		CLA	BoEA MIC	BoEA 2 × MIC	BoEA 4 × MIC	ElAc MIC	ElAc 2 × MIC	ElAc 4 × MIC
0	5.9 ± 0.08	5.9 ± 0.08	5.9 ± 0.08 **	5.9 ± 0.08 **	5.9 ± 0.08 **	5.9 ± 0.08 **	5.9 ± 0.08 **	5.9 ± 0.08 **
12	7.03 ± 0.12	6.76 ± 0.16	6.83 ± 0.12 **	6.83 ± 0.12 **	6.9 ± 0.08 **	6.83 ± 0.12 **	6.66 ± 0.23 **	5.66 ± 0.12 **
24	8.9 ± 0.08	6.73 ± 0.16	7.23 ± 0.16 **	7.23 ± 0.16 **	6.53 ± 0.12 **	6.3 ± 0.08 **	6.06 ± 0.09 **	5.76 ± 0.26 **
36	9.1 ± 0.16	6.9 ± 0.08	7.4 ± 0.37 **	7.1 ± 0.08 **	6.33 ± 0.12 **	6.53 ± 0.12 **	5.83 ± 0.16 **	5.73 ± 0.20 **
48	9.76 ± 0.04	7.03 ± 0.12	8.5 ± 0.40 *	7.86 ± 0.04 *	6.66 ± 0.12 *	6.8 ± 0.08 **	5.76 ± 0.04 **	5.30 ± 0.08 **
72	10.26 ± 0.44	7.5 ± 0.21	8.5 ± 0.37 *	7.23 ± 0.30 *	6.66 ± 0.12 *	7.0 ± 0.08 **	6.23 ± 0.16 **	5.36 ± 0.26 **
96	10.7 ± 0.16	8.1 ± 0.14	9.7 ± 0.24 *	8.46 ± 0.41 *	7.16 ± 0,12 *	7.36 ± 0.09 **	6.23 ± 0.30 **	5.33 ± 0.20 **
120	9.7 ± 0.16	8.33 ± 0.12	9.9 ± 0.08 *	9.1 ± 0.08 *	8.33 ± 0.23 *	6.86 ± 0.12 **	7.03 ± 0.12 **	6.26 ± 0.18 **

Data are presented as the mean ± standard deviation (SD) of Colony-Forming Unit (CFU) and representing of one of three independent experiments performed in triplicate. Statistical analysis was performed using analysis of variance and post hoc t-test. * *p* < 0.05, ** *p* < 0.005 when compared to the growth control. Abbreviations: BoEA, ethyl acetate fraction of *B. orellana*; E1Ac, ellagic acid; MIC, minimal inhibitory concentration.

## Data Availability

Not applicable.

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
