# Peer review of "Ethyl Acetate Fraction of Bixa orellana and Its Component Ellagic Acid Exert Antibacterial and Anti-Inflammatory Properties against Mycobacterium abscessus subsp. massiliense"

_antibiotics, 2022, doi:10.3390/antibiotics11060817_

Round 1

Reviewer 1 Report

The article Ethyl Acetate Fraction of Bixa orellana and Its Component Ellagic Acid Exert Antibacterial and Anti-inflammatory proper-3 ties against Mycobacterium abscessus subsp. massiliense after consideration of the following  comments

.

  • Abstract, authors should be mentioned standard results for both antibacterial and anti-inflammatory..
  • Also it is known that ellagic acid has anti-inflammatory activity what is the addition to that result from this study .
  • Introduction, the rational for this study should be improved.
  • Also, the literature studies mentioned that the chemical composition of the extracts and authors mentioned that ellagic acid is the main component

So authors should use authentic ellagic acid dissolved in ethyl acetate as standard in all assays.

  • It is better for authors to add natural FDA approved contain ellagic acid or related derivatives.
  • I think the results of this study can be reliable if the authors determine the concentration of ellagic acid in the extracts.
  • Also, using of mice in evaluation of anti-inflammatory activity is not reliable.
  • I suggest supporting this study with some docking studies to predict the main target of ellagic acid
  • Conclusions need to improve.

Reviewer 2 Report

Manuscript Number: antibiotics-1741806

The manuscript entitled: Ethyl Acetate Fraction of Bixa orellana and Its Component El-2 lagic Acid Exert Antibacterial and Anti-inflammatory proper-3 ties against Mycobacterium abscessus subsp. Massiliense by Moraes-Neto RN et al., investigated the biological activities of the ethyl acetate fraction of BoEA and ElAc, focusing on antimicrobial activity against Mycobacterium abscessus subsp. Massiliense.

The manuscript needs to be really improved, mainly by detailing/enhancing the text as follows.

Introduction section

ü  Lines 43-45: insert some references for this sentence and indicate the percentages of infection on FC patients.

ü  Line 47: detail the intrinsic mechanisms of resistance of Mabs.

ü  Lines 48-49: delete the sentence “clique ou toque……”.

ü  Line 65: insert a reference for this statement.

Results section

ü  Lines 133-137: change the presentation mode of this part of the results, table 6 is not necessary.

ü  Lines 147-151: the results of the time kill have to be presented in a table to better quantify (as CFU numbers) the obtained reduction.

ü  Lines 177-181: the cytokine release pattern involved in the edema reduction should be presented.

Discussion section

The discussion section is very poor in contents and too generic. The authors report few comparisons with literature data. They should rewrite it to improve its overall quality, and by adding the limitations of the study.

Material and Methods section

Lines 394-398: a reference Mabs strain should be used as control for this part of the study.

Conclusion section

This section, once again, is too generic. The authors should at least report the aspects that should be studied further.

Reference section

The references should be write according to the journal instructions.

Round 2

Reviewer 2 Report

Manuscript Number: antibiotics-1741806

The manuscript entitled: Ethyl Acetate Fraction of Bixa orellana and Its Component El-2 lagic Acid Exert Antibacterial and Anti-inflammatory proper-3 ties against Mycobacterium abscessus subsp. Massiliense by Moraes-Neto RN et al., investigated the biological activities of the ethyl acetate fraction of BoEA and ElAc, focusing on antimicrobial activity against Mycobacterium abscessus subsp. Massiliense.

 The authors made a relevant effort to improve the overall quality of the paper since many modifications were made in the manuscript text, addressing further comments in the letter.

In any case, the reviewer considers it necessary for the authors to implement the English of the text.